# The Antibacterial Activity Mode of Action of Plantaricin YKX against *Staphylococcus aureus*

**DOI:** 10.3390/molecules27134280

**Published:** 2022-07-03

**Authors:** Jinjin Pei, Yigang Huang, Ting Ren, Yaodong Guo, Jun Dang, Yanduo Tao, Yonggui Zhang, A. M. Abd El-Aty

**Affiliations:** 1Northwest Institute of Plateau Biology, Chinese Acadamy of Science, Xining 530001, China; jinjinpei@snut.edu.cn (J.P.); dangjun@nwipb.cas.cn (J.D.); 2Shaanxi Province Key Laboratory of Bioresources, 2011 QinLing-Bashan Mountains Bioresources Comprehensive Development C. I. C., Qinba State Key Laboratory of Biological Resources and Ecological Environment, College of Bioscience and Bioengineering, Shaanxi University of Technology, Hanzhong 723001, China; hyg429397653@163.com (Y.H.); rtening@163.com (T.R.); 3College of Health and Management, Shangluo University, Shangluo 724900, China; yaodongguo@163.com; 4Department of Pharmacology, Faculty of Veterinary Medicine, Cairo University, Giza 12211, Egypt; 5Department of Medical Pharmacology, Faculty of Medicine, Atatürk University, Erzurum 25240, Turkey

**Keywords:** *S. aureus*, plantaricin YKX, antibacterial effect, biofilm

## Abstract

We aimed to evaluate the inhibitory effect and mechanism of plantaricin YKX on *S. aureus*. The mode of action of plantaricin YKX against the cells of *S. aureus* indicated that plantaricin YKX was able to cause the leakage of cellular content and damage the structure of the cell membranes. Additionally, plantaricin YKX was also able to inhibit the formation of *S. aureus* biofilms. As the concentration of plantaricin YKX reached 3/4 MIC, the percentage of biofilm formation inhibition was over 50%. Fluorescent dye labeling combined with fluorescence microscopy confirmed the results. Finally, the effect of plantaricin YKX on the AI-2/LuxS QS system was investigated. Molecular docking predicted that the binding energy of AI-2 and plantaricin YKX was −4.7 kcal/mol and the binding energy of bacteriocin and luxS protein was −183.701 kcal/mol. The expression of the *luxS* gene increased significantly after being cocultured with plantaricin YKX, suggesting that plantaricin YKX can affect the QS system of *S. aureus*.

## 1. Introduction

*Staphylococcus aureus* (*S. aureus*), one of the leading pathogenic bacteria, is able to form biofilms and can contaminate foods and medical equipment. Its biofilms are observed in processing equipment and open surfaces, resulting in disease outbreaks [1,2,3,4,5]. Compared with free cells, biofilms can not only enhance the resistance of bacteria to unpleasant environments (ultraviolet light, acid, and alkali) but also enhance the resistance of bacteria to the host immune defense system [6,7,8,9].

Chemical preservatives have often been harmful to human health [10,11,12]. Lactobacteriocin (Bactoriocins) is a class of peptide or peptide complexes produced by lactic acid bacteria (LAB) with the ability to inhibit or kill other forms similar to their proximity or living environment [13,14]. It is generally accepted as naturally nontoxic [15,16]. Many researchers have focused on the application of bacteriocins. Nisin is the first bacteriocin approved for use in food. At present, more than 45 countries have recognized nisin as a safe food preservative. As the problem of antibiotic resistance is becoming increasingly serious, people are constantly looking for drugs that can replace antibiotics against drug-resistant bacteria, among which bacteriocin is considered to have great potential. In the production of animal husbandry, with the emergence of bacteriocin and the deepening of research, it is gradually possible for bacteriocin to replace antibiotics to ensure the health of livestock and poultry, the safety of animal products and the safety of the breeding ecological environment. Bacteriocin is also often used in conjunction with other treatments and can be used as a fence to improve food safety. Regarding the study of the antibacterial activity mechanism of lactic acid bacteriocins, the literature has focused on the inhibition/killing mechanism of sensitive bacteria [10,11]. Bacteriocins can form hydrophobic “holes” in the cell membrane of sensitive bacteria, resulting in loss of cell content and cell death [17]. Furthermore, they can combine with vital proteins (enzymes) and DNA within sensitive bacteria, causing cell death [18]. Recently, some studies have found that some lactic acid bacteria can similarly affect biofilm formation in sensitive bacterial organisms [19,20]. However, a few studies have investigated the effect of bacteriocin on the quorum sensing (QS) system of sensitive bacteria.

Biofilms are composed of multilayer bacteria and extracellular polymers secreted by them. Bacteria in biofilms regulate the expression of different genes in bacteria through signal transduction between bacteria [2,3]. The bacteria in different parts of the biofilm play different physiological functions and produce synergy to maintain the spatial structure and function of the biofilm and make the biofilm become a whole with multicellular function [5]. This regulatory signaling molecule between bacteria is synthesized by the quorum sensing system, which is called autoinducer (AI) [4]. The quorum sensing system is common in bacteria. Different strains have similar regulation methods and functions. The quorum sensing system can regulate the expression of bacteria-related genes in biofilms and control the growth mode [5,6]. It is also related to the formation of biofilms. Therefore, the important role of the quorum sensing system in bacterial biofilm formation makes it a hotspot in the study of biofilm formation mechanisms.

Class IIa LAB bacteriocin plantaricin YKX (Lys-Tyr-Gly-Asn-Gly-Leu-Ser-Arg-Ile-Phe-Ser-Ala-Leu-Lys) was previously isolated, purified, and identified as a novel bacteriocin. It was found to be able to kill *S. aureus* and prevent biofilm formation [21]. This study aimed to investigate the mechanism of action of plantaricin YKX on *S. aureus* and its biofilm formation by investigating the effect of plantaricin YKX on the AI-2/LuxS QS system. It is of scientific significance to provide new perspectives and ideas for *S. aureus* control research and the antibacterial activity of lactic acid bacteria.

## 2. Materials and Methods

### 2.1. Materials and Strains

Plantaricin YKX was produced by *Lactobacillus plantarum* YKX, as described in a previous study [21]. *S. aureus* strains were used in this study because they are proficient in forming biofilms. LB broth (Solarbio, Beijing, China) was used to cultivate *S. aureus,* and MRS broth (Solarbio, Beijing, China) was used for the *L. plantarum* culture.

### 2.2. Determination of the Minimum Inhibitory Concentration (MIC)

The minimum inhibitory concentration was determined according to Clinical and Laboratory Standards (Clinical and Laboratory Standards Institute, 2012) [22]. The mid-logarithmic phase of *S. aureus* strains (listed in Table 1) was cocultured with planttaricin YKX at final concentrations of 256 μg/mL, 128 μg/mL, 64 μg/mL, 32 μg/mL, 16 μg/mL, 8 μg/mL, and 0 μg/mL. After 24 h of culturing at 37 °C, the first diluted concentration where *S. aureus* cell growth was inhibited was defined as the minimum inhibitory concentration (MIC).

### 2.3. Effect of Plantaricin YKX on S. aureus at Different Growth Phases

The effect of plantaricin YKX on the cells of *S. aureus* ATCC 25923 and *S. aureus* CMCC 26570 at different growth phases was investigated according to our previous work (Pei et al., 2013). Plantaricin YKX was added to a culture of *S. aureus* at early, middle exponential, and early stationary phases, respectively, with a final concentration of 2 MIC. Viable cell numbers were recorded every 5 h by plating onto LB agar.

### 2.4. Effect of Plantaricin YKX on S. aureus Cells

#### 2.4.1. Measurement of the Extracellular K^+^ Ion Concentration

The leakage of K^+^ ions from the cells of *S. aureus* grown at different pH values after treatment with plantaricin YKX was determined according to Pei et al., 2013 [23]. Planticicin YKX was added to the mid-exponential phase of *S. aureus* grown at pH 6, pH 7, and pH 8. The change in K^+^_out_ was recorded over a period of one hour with a PHG potassium ion selective electrode (Van London-pHoenix, Houston, TX, USA). The level of K^+^_in_ was calculated based on the formula (K^+^_in_ = K^+^_total_ − K^+^_out_). Total intracellular K^+^ (K^+^_total_) was determined by treating cells with 10 mg/mL lysozyme at 37 °C for 30 min and determining the K^+^.

#### 2.4.2. Flow Cytometry

Flow cytometry was conducted according to Pei et al., 2018 [24]. *S. aureus* ATCC 25,923 bacterial cells were incubated with plantaricin YKX at 1× MIC (control: cells treated without plantaricin YKX) for 1 h at 37 °C. The samples were then stained with propidium iodide (10 µg/mL) for 15 min at 37 °C. Fluorescence intensity was analyzed using an Accuri C6 flow cytometer (BD Biosciences, Ann Arbor, MI, USA) with an emission wavelength of 617 nm and an excitation wavelength of 535 nm.

#### 2.4.3. Scanning Electron Microscope Observation of *E. coli* Cells

SEM was conducted according to Pei et al., 2018 [24]. Mid-logarithmic-phase *S. aureus* ATCC 25923 was treated with plantaricin YKX at a final concentration of 1 MIC or 2 MIC for 1 h. The samples without plantaricin YKX treatment were considered controls. After fixation with 2.5% glutaraldehyde solution, the samples were dehydrated with a serial concentration of ethanol. Then, the samples were observed with an SU-8010 field emission SEM (Hitachi High New SU8010 Series, Hitachi High-Tech Co., Ltd., Tokyo, Japan).

### 2.5. Effect of Plantaricin YKX on the Biofilm of S. aureus

#### 2.5.1. Inhibitory Activity of Plantaricin YKX on *S. aureus* Biofilm Formation

Crystal violet staining was used to determine the inhibition of the biofilm of *S. aureus* according to Pei et al., 2021 [21]. Mid-logarithmic-phase *S. aureus* (200 μL) was added to 96-well plates. Plantaricin YKX solution was added at final concentrations of 1/4 MIC, 1/2 MIC, and 3/4 MIC (the sub-MIC concentration of plantaricin YKX was used to eliminate the possibility that the inhibition of biofilm formation was due to the cells being killed by plantaricin YKX). The isolate was cultured at 37 °C for 3 days for biofilm formation. The samples without plantaaricin YKX treatment were used as controls. The biofilms were fixed with methanol and then stained with crystal violet (2% concentration). After dropping the staining agents, 95% ethanol solution was added and shaken (100 rpm/min) for 30 min. The OD value of the supernatant (100 μL) was tested at 595 nm.

#### 2.5.2. Confocal Laser Microscopy

Mid-logarithmic-phase *S. aureus* ATCC 25923 was added to plantaricin YKX at a final concentration of 1/2 MIC. The isolate was cultured at 37 °C for 3 days for biofilm formation. The samples without plantaaricin YKX treatment were used as controls. According to the manufacturer’s instructions [25], a LIVE/DEAD BacLight Bacterial Viability Kit/SYTO9/PI dye (Cell Viability, Proliferation & Function) was used to stain the samples for 30 min. Confocal laser microscopy (1000× oil lens) was used to photo the samples with a SYTO 9 excitation wavelength of 480 nm, an emission wavelength of 500 nm, a PI excitation wavelength of 490 nm, and an emission wavelength of 635 nm.

### 2.6. Effect of Plantaricin YKX on the QS System of S. aureus

#### 2.6.1. Molecular Docking Was Used to Investigate the Effect of Plantaricin YKX on AI-2 Molecules and the Lux S Enzyme

Chembiodraw Ultra 14.0 (Cambridge Company, New York, NY, USA) was used to draw Al-2 of *S. aureus*, and Chembio3D Ultra 14.0 was used for energy minimization to set the minimum RMS gradient to 0.001 (the molecule was saved in MOL2 format). The optimized molecule was imported into AutoDockTools-1.5.6 (National Institute of Health, NIH, Bethesda, MA, USA) for hydrogenation, charge calculation, and charge distribution, and the rotatable key was set and saved in “PDBQT” format. PyMOL constructed the structure of plantaricin YKX. AutoDock Vina1.1.2 was used to dock the small molecule Al-2 with a polypeptide (plantaricin YKX), and the relevant parameters were set as follows: center_x = −6.704, center_y = 38.513, and center_z = 3.778 as central coordinates. The grid box size is 40 × 40 × 40 (the spacing of each grid is 0.375 A), and the remaining parameters are set as default.

#### 2.6.2. Effect of Plantaricin YKX on the mRNA Expression of QS System-Related Genes (*ica*, *lux* and *hld*)

To verify the effect of plantaricin YKX on the QS system, RT–qPCR was used to detect the mRNA expression changes in the *icaC, lux,* and *hld* genes of *S. aureus.* According to the sequences of *ica, lux,* and *hld* in Geneback, primers (listed below as Table 2) were designed with Primer 5.0 software and synthesized by Shengon Bioengineering Lt. Co.Shanghai, China The mid-logarithmic phase cells of *S. aureus* ATCC 25923 were treated with plantaricin YKX at 1/4 MIC, 1/2 MIC, and 3/4 MIC. After overnight incubation, RNA was extracted according to the TRIzol method.

Total RNA 2 μL; DNTP 0.5 μL; random primer 0.5 μL; and 4 μL of distilled water without RNA enzyme were used to form a reverse transcription reaction system (10 μL). All samples were kept at 70 °C for 5 min and then transferred to an ice bath for preservation. Immediately added were 2 μL of 5× reverse transcription buffer, 0.5 μL of RNA enzyme inhibitor, and 0.5 μL of MMLV reverse transcriptase. Reverse transcription was performed at 30 °C for 10 min, 42 °C for 1 h, 70 °C for 15 min, and 4 °C indefinitely. After reverse transcription, the DNA template was diluted 5 times and stored at −20 °C until use.

Upstream primer (10 μM) 0.2 μL, downstream primer (10 μM) 0.2 μL, 2× Ultra SYBR Mixture 5 μL, ddH_2_O 3.6 μL, and DNA template 1 μL were then used for PCR amplification. 16S rRNA was used as an internal standard. The reaction parameters were 95 °C for 10 min, 95 °C for 15 s, and 60 °C for 1 min, and the cycle was 40 times.

### 2.7. Statistical Analyses

Data are expressed as the mean ± SD. One-way ANOVA was carried out using SPSS 18.0 (IBM, Armonk, New York, NY, USA) software, and *p* < 0.05 was considered statistically significant.

## 3. Results and Discussion

### 3.1. The MIC of Plantaricin YKX

The living cell number of *S. aureus* decreased with increasing plantaricin YMX concentration. As shown in Table 1, the minimum inhibitory concentration of plantaricin YMX against *S. aureus* was 16–32 μg/mL.

Generally, the MICs of bacteriocins against *S. aureus* are low [11,20]. The MIC is 16 μg/mL plantaricin YKX against *S. aureus,* which is similar to the MIC of bactreiocin SLG10 [26]. However, there are also some bacteriocins with low MICs against *S. aureus*; for example, the MIC of bacteriocin SLG1 is only 8 μg/mL [24]. The antibacterial activity of bacteriocins might be different with different strains. Similar results have also been found for bacteriocin Ent35-MccV [25], bacteriocin C2 [11], and bacteriocin SLG1 [24].

### 3.2. Effect of Plantaricin YKX on S. aureus Cells at Different Growth Phases

The addition of plantaricin YKX to an early exponential phase culture of *S. aureus* repressed cell growth (Figure 1). Similar inhibition was observed when plantaricin YKX was added to the mid-exponential and stationary phase cultures (Figure 1). The inhibitory effect of plantaricin YKX was not affected by the difference in the strains of *S. aureus*.

The growth phases of sensitive bacteria might affect the resistance to bacteriocins because in different growth phases, there might be differences in the detailed compositions of cell membranes. Several bacteriocins reported that the antibacterial activity exhibited differences according to the different growth phases of the sensitive bacteria [27]. However, similar to plantaricin YKX in this study, many bacteriocin antibacterial abilities were not affected by the growth phases of sensitive bacteria, such as bacteriocin SLG10 [26], bacteriocin Ent35-MccV [25], and plantaricin K25 [18].

### 3.3. Effect of Plantaricin YKX on S. aureus Cells

After the addition of plantaricin YKX, K^+^ was immediately and rapidly effluxed, followed by a later slow release (Figure 2). With the addition of plantaricin YKX, cells incubated at pH 8.0 lost the least K^+^ during 1 h (Figure 2), and cells incubated at pH 7.0 lost the lowest K^+^ concentration for the next hour. The efflux rate decreased significantly as the extracellular pH dropped to 6 (Figure 2).

The percentage of dead bacteria in the control group was 0.24%, while it was 11.5% for the ATCC 25923 strain and 12.4% for CMCC 26570 after treatment with plantaricin YKX (Figure 3). The results indicated that plantaricin YKX has antibacterial activity against *S. aureus* ATCC.

As shown in Figure 4, compared to the untreated *S. aureus* cells, the cells treated with plantaricin YKX showed obvious damage. SEM images indicate that plantaricin YKX might form pores on cell membranes, leading to the release of cytoplasmic components. The SEM observation results are consistent with the leakage of potassium testing and the flow cytometry test.

Our results suggest that plantaricin YKX acts on the cytoplasmic membrane of *S. aureus*. The generation of antibacterial activity may be due to the creation of pores in the cell membrane. The observation of K^+^ loss showed that pH is one of the important factors affecting the ability of plantaricin YKX to form pores in *S. aureus* cell membranes. The highest rate of bacteriocin-induced K^+^ loss was observed at pH 6. As observed for several other bacteriocins, ΔpH may contribute to the action of bacteriocins (under physiological conditions) [23,27].

Flow cytometry analysis revealed that plantaricin YKX mediated cell membrane destruction and eventually led to cell death. Similar results were reported for nisin, plantaricin K25 [28], bifidocin A [18], and enterocin 7A and 7B [29].

Based on the current literature, we know that cell membrane permeability plays an important role in the mode of action of bacteriocins. In this study, plantaricin YKX led to pore formation and eventual cell death, demonstrating the ability to increase cell membrane permeability. Similar mechanisms were observed with plantaricin 163 [30], bifidocin A, and bacteriocin Ent35-MccV [25,29].

### 3.4. Inhibition of Biofilm Formation by Plantaricin YKX

As shown in Figure 5, the sub-MIC concentration of plantaricin YKX inhibited the formation of biofilms of *S. aureus* ATCC 25923, and the rates of plantaricin YKX inhibition of *S. aureus* biofilm formation reached 16.3%, 46.6%, and 56.1% at 0.25× MIC, 0.5× MIC, and 0.75× MIC, respectively.

Fluorescent probe staining was used for the observation and measurement of bacterial adhesion. As shown in Figure 6, there were fewer biofilm adhesion bacteria in the presence of plantaricin YKX. The bacterial colonies in the control were dense, and the number of live bacteria within the biofilm was obviously higher than those treated with plantaricin YKX.

It is hypothesized that 99% of bacteria can exist in the form of biofilms [31]. Biofilm formation is also one of the important factors contributing to food contamination, and plantaricin YKX may provide a new idea to prevent food contamination by pathogens or spoilage bacteria by inhibiting biofilm formation. Previous studies on the mode of action of lactic acid bacteria bacteriocins have focused on the effect of bacteriocins on sensitive bacterial cells, and some recent studies have suggested that bacteriocins may also affect biofilm formation in sensitive bacteria [20,31].

### 3.5. Effect of Plantaricin YKX on the AI-2/Lux S QS System of S. aureus

The binding energy of AI-2 and plantaricin YKX is −4.7 kcal/mol, which proves that Al-2 has a good binding effect. Al-2 interacts with plantaricin YKX mainly through hydrogen bonding and forms a hydrogen bond with S7 with an A length of 3.0 Å (Figure 7).

The expression of the *luxS* and *icaC* genes of *S. aureus* ATCC 25923 increased significantly (*p* < 0.05) after being cocultured with plantaricin YKX at 1/4× MIC, 1/2× MIC, and 3/4× MIC (Figure 8). However, the expression of the *hld* genes of *S. aureus* ATCC 25923 after plantaricin YKX treatment was not changed significantly (*p* > 0.05) (Figure 8C).

QS systems are one of the research hotspots in the field of microbiology. It can be divided into three types: (1) LuxS/Al-2-dependent QS systems; (2) oligopeptide-mediated two-component sensing systems in Gram-positive bacteria; and (3) AHL lux I/Lux R systems in Gram-negative bacteria [7]. QS systems are related to bacterial species in many-to-one manners. For example, LuxS/AI-2-dependent QS systems are present in both Gram-positive and Gram-negative bacteria, where AI-2 is a universal signaling molecule involved in information exchange [4]. Due to bacterial swarming, the LuxS/AI-2-dependent QS system and AI-2 molecules are essential for bacteria to form a relatively stable ecological environment for bacterial population and functional division. It is closely linked to biological functions such as bacteriocin synthesis, biofilm formation, and virulence factor expression.

In the initial stage of biofilm formation, polysaccharide intercellular adhesin (PIA) encoded by the icaadb operon is an important mediator in the aggregation stage [7]. The formation of biofilms depends on the expression of the *ica* gene and PIA synthesis [4]. Staphylococcus has a unique and complete system. The *arg* and *luxS* genes are the key genes in the regulation mechanism of QS [6]. The *agr* gene is responsible for the dependent regulation of the growth stage of virulence factors. The *agr* system can regulate the adhesion of the bacterial surface. The *agr* system consists of two different transcription units, RNA II and RNA III. RNA III is also the messenger RNA of the *hld* gene, which encodes D toxin [8]. Another QS system of *Staphylococcus*, the LuxS system, can downregulate PIA products, an important mediator in the aggregation stage [32]. Therefore, the QS system of *Staphylococcus* plays an important role in the biofilm formation of bacteria.

All AI-2 is a byproduct of the methyl cycle. LuxS protein, a key and essential enzyme for Al-2 synthesis, plays an important role in the methyl cycle [4]. We evaluated the effect of plantaricin YKX on *icaC, luxS,* and *hld* gene expression. The results indicate that plantaricin YKX has a positive effect on *icaC* and *luxS* gene expression. Previous studies on the mode of action of bacteriocins have focused on the mechanism of bacteriocins on susceptible bacteria [18,29], while little has been reported on the effect on the QS system of susceptible bacteria. The results of the present study suggest that the conclusion of the effect on the QS system of susceptible bacteria does not hold for all bacteriocins; plantaricin YKX does, but nisin does not [21]. Thus, the mode of action of bacteriocins seems to be diverse. Similar reports have been mentioned, for example, for plantaricin K25 [18], bifidocin A [29], plantaricin 163 [30], and plantaricin JLA-9 [33].

## 4. Conclusions

In this study, the inhibitory mode of action of plantaricin YKX on *S. aureus* was investigated. The MIC of plantaricin YKX against *S. aureus* was 16 μg/mL. After treatment with sub-MIC plantaricin YKX, living cells in the biofilm of *S. aureus* were significantly decreased. Confocal laser microscopy showed that 1/2 MIC plantaricin YKX was able to inhibit the formation of biofilms. Molecular docking suggested that plantaricin YKX was able to bind with AI-2 molecules and its synthesis enzyme Lux S. Additionally, plantaricin YKX was also able to affect the gene expression of the QS system-related gene. Through the results, the conclusion can be obtained that plantaricin YKX was not only able to damage the cell membrane of *S. aureus* but also able to regulate its QS system.

## Figures and Tables

**Figure 1 molecules-27-04280-f001:**
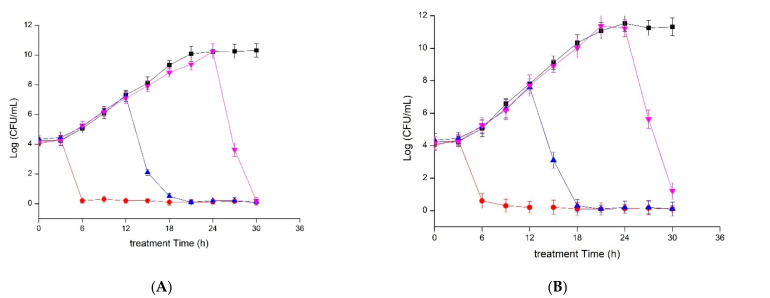
Inhibitory kinetics of plantaricin YKX against *S. aureus* ATCC 25923 (**A**) and *S. aureus* CMCC 26570 (**B**). (*■*): Control (without plantaricin YKX treatment); plantaricin YKX was added at the early-stage phase (●), at the mid-exponential phase (*▲*), and the stationary phase (*▼*).

**Figure 2 molecules-27-04280-f002:**
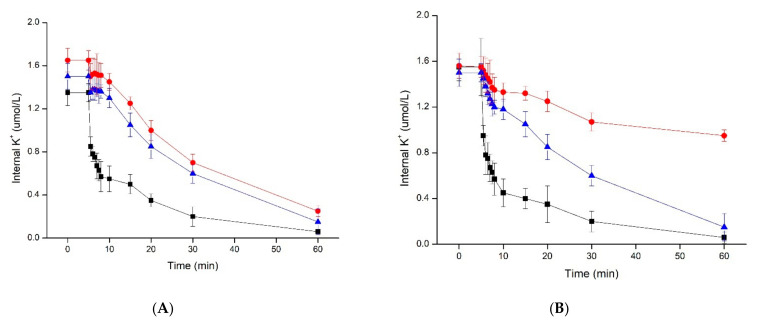
Efflux of K^+^ from cells of *S. aureus* ATCC 25923 (**A**) and *S. aureus* CMCC 26570 (**B**) after treatment with plantaricin YKX at pH 8 (●), pH 7 (▲), and pH 6 (■).

**Figure 3 molecules-27-04280-f003:**
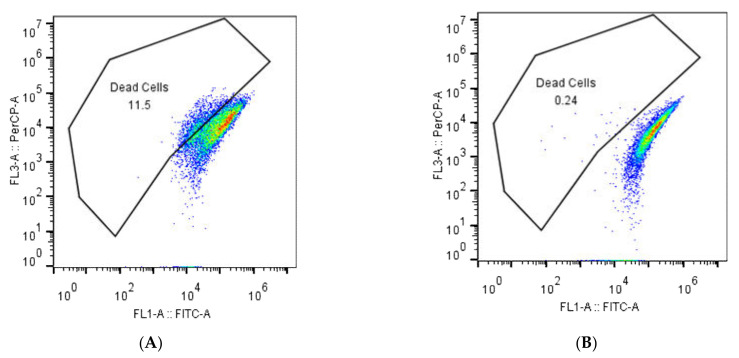
Flow cytometry results of *S. aureus* ATCC 25923 without (**A**) and with (**B**) plantaricin YKX treatment.

**Figure 4 molecules-27-04280-f004:**
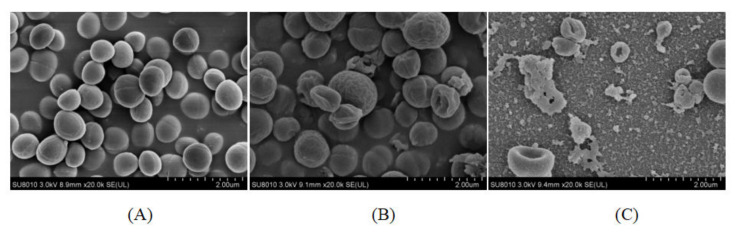
Effect of plantaricin YKX on the cell structure of *S. aureus* ATCC 25923. (**A**): without plantaricin YKX treatment; (**B**): after treatment with 1 MIC plantaricin YKX for 30 min; (**C**): after treatment with 1 MIC plantaricin YKX for 2 h.

**Figure 5 molecules-27-04280-f005:**
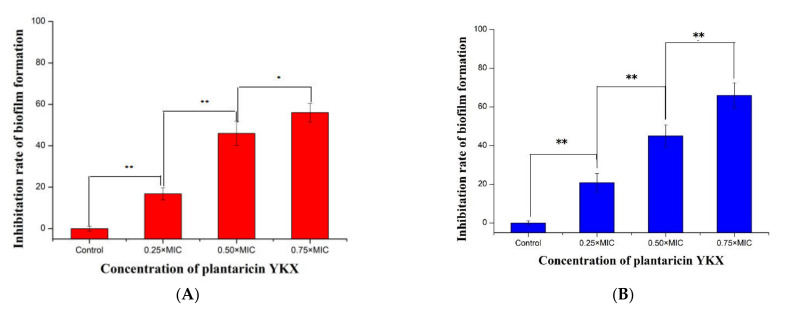
Effect of plantaricin YKX on *S. aureus* biofilm formation. *S. aureus* ATCC 25923 (**A**) and *S. aureus* CMCC 26570 (**B**). * significant. *p* < 0.05; ** highly significant. *p* < 0.01.

**Figure 6 molecules-27-04280-f006:**
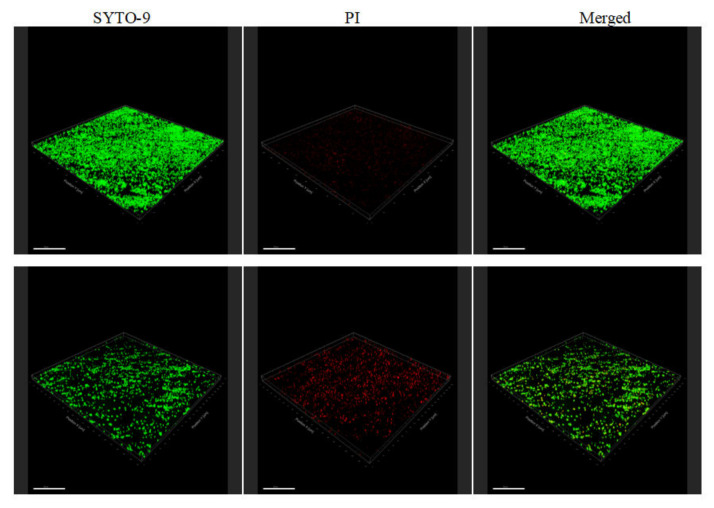
Confocal laser observation of biofilms of *S. aureus* treated without (**Up**) and with (**Down**) 3/4 MIC plantaricin YKX.

**Figure 7 molecules-27-04280-f007:**
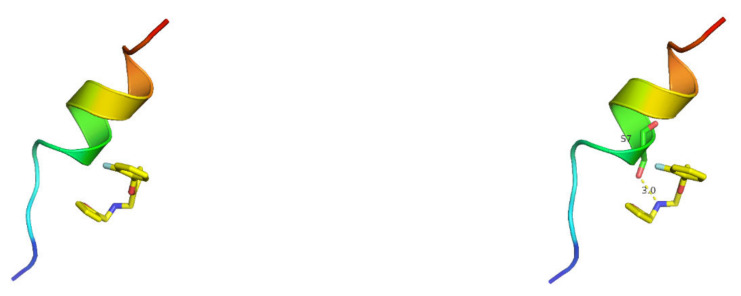
Docking results of plantaricin YKX with Al-2 of *S. aureus*. (**Left**): homology modeling results. (**Right**): docking results.

**Figure 8 molecules-27-04280-f008:**
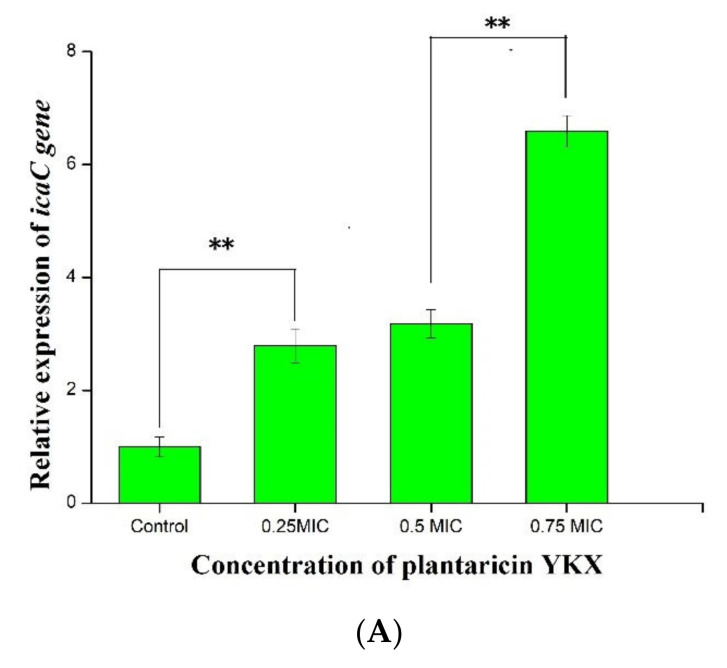
Effect of plantaricin YKX on the relative expression of *QS system-related* genes. (**A**): *icaC*; (**B**): *hld*; (**C**): *luxS*. * significant. *p* < 0.05; ** highly significant. *p* < 0.01.

**Table 1 molecules-27-04280-t001:** MIC of plantaricin YKX against *S. aureus*.

Strains of *S. aureus*	MIC
*S. aureus* CMCC 26560	16 μg/mL
*S. aureus* CMCC 26563	16 μg/mL
*S.aureus* CMCC 26565	16 μg/mL
*S. aureus* CMCC 26570	16 μg/mL
*S. aureus* CMCC 26573	32 μg/mL
*S. aureus* CMCC 26581	16 μg/mL
*S. aureus* CMCC 26590	32 μg/mL
*S. aureus* CMCC 26575	16 μg/mL
*S.aureus* CMCC 26585	16 μg/mL
*S. aureus* CMCC 26579	16 μg/mL
*S. qureus* ATCC 25923	16 μg/mL

**Table 2 molecules-27-04280-t002:** Primers used in this study.

Gene	Primers (5′-3′)
*lux*	F: *TCCTATGGGTTGTCAAACTGG*
	R: *CCTTCTCCGTAGATGTCATTCC*
*icaC*	F: *TGCTTACACCAACATATTTGAAGATAATAC*
	R: *GACGCCTATACAAATTCCTAGAATCATT*
*hld*	F: *GAAGTTATGATGGCAGCAGAT*
	R: *GTTGGGATGGCTCAACAACT*
*16S rRNA*	F: *GCGGTCGCCTCCTAAAAG* R: *TCCCGGTCCTCTCGTACTA*

## Data Availability

All the data are included in this manuscript.

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
