# Peer review of "The Antibacterial Activity Mode of Action of Plantaricin YKX against Staphylococcus aureus"

_molecules, 2022, doi:10.3390/molecules27134280_

Round 1

Reviewer 1 Report

The manuscript focuses on the anti-bacterial activity of the peptide plantaricin YKX. Even though the characterization of the bacteriocins lacks novelty, the manuscript is well designed, at least its first part. After the bactericidal analysis of the peptide, the paper deals with the changes in the membrane permeability induced by this plantaricin. The authors can study this process by recording the leakage of potassium ions, the changes in the structure by SEM and also by analyzing the uptake of propidium iodide using a flow cytometer. Of note, the excitation and emission wavelengths used for propidium are not correct (section 2.4.2).

The second part includes the characterization of the biofilm formation of Staphylococcus aureus and how the plantaricin can interfere with that. By the way, what is the difference between Fig. 5A and 5B? Beyond that, there is a good approach with confocal microscopy, and an interesting analysis of the expression of key genes involved in biofilm formation, but then there is somehow troublesome in silico analysis of the interaction of plantaricin YKX and the autoinducer-2 and the LuxS. This protein is intracellular; thus, it is highly unlikely that plantaricin YKX interacts with LuxS unless the cell is completely broken and dead. Besides, when analyzing the interaction with the autoinducer-2 and LuxS, a folded peptide is used, i.e. with a helix formed. This is another unlikely event. It is generally accepted that peptides have no structure when present in aqueous media such as the extracellular milieu or the bacterial cytosol. These peptides can form secondary structures in the present of membranes or membrane-like environments. Therefore, the in silico approach should be thoroughly revised.

Author Response

Responses to Comments for reviewer 1

The manuscript focuses on the anti-bacterial activity of the peptide plantaricin YKX. Even though the characterization of the bacteriocins lacks novelty, the manuscript is well designed, at least its first part. After the bactericidal analysis of the peptide, the paper deals with the changes in the membrane permeability induced by this plantaricin. The authors can study this process by recording the leakage of potassium ions, the changes in the structure by SEM and also by analyzing the uptake of propidium iodide using a flow cytometer. Of note, the excitation and emission wavelengths used for propidium are not correct (section 2.4.2).

Answer: I am very sorry for such mistakes. flow cytometer were tested by Zhinanzhen Company and the information were confused by our. The excitation and emission wavelengths used for propidium are corrected in manuscript  (L92-93).

The second part includes the characterization of the biofilm formation of Staphylococcus aureus and how the plantaricin can interfere with that. By the way, what is the difference between Fig. 5A and 5B?

Answer:Sorry for the confusion. Fig.5A is the for Strain S. aureus ATCC 25923 and 5B is for S. aureus CMCC 26570. 

Beyond that, there is a good approach with confocal microscopy, and an interesting analysis of the expression of key genes involved in biofilm formation, but then there is somehow troublesome in silico analysis of the interaction of plantaricin YKX and the autoinducer-2 and the LuxS. This protein is intracellular; thus, it is highly unlikely that plantaricin YKX interacts with LuxS unless the cell is completely broken and dead. Besides, when analyzing the interaction with the autoinducer-2 and LuxS, a folded peptide is used, i.e. with a helix formed. This is another unlikely event. It is generally accepted that peptides have no structure when present in aqueous media such as the extracellular milieu or the bacterial cytosol. These peptides can form secondary structures in the present of membranes or membrane-like environments. Therefore, the in silico approach should be thoroughly revised.

Answer: thank you very much for the comments. Yes, Lux S is intracellular. So we deleted the silico analysis of plantaricin YKX and the Lux S enzyme from this manuscript according to the reviewer’s suggestions.

Reviewer 2 Report

The aim of the study was to investigate the antibacterial activity mode of action of plantaricin YKX 2 against Staphylococcus aureus. Although it is an interesting and well-designed work and significant results are presented, it is of limited novelty to my opinion, since several similar studies are available in the literature.

It is essential that the authors highlight the innovative elements of their work.

Other points

To my opinion, an introductory paragraph on bacteriocins (activity, applications, etc) should be included in the Introduction.

Discussion of potential application of the plantaricin is missing. Has it been tested in real systems?

Author Response

Responses to Comments for reviewer 2:

The aim of the study was to investigate the antibacterial activity mode of action of plantaricin YKX 2 against Staphylococcus aureus. Although it is an interesting and well-designed work and significant results are presented, it is of limited novelty to my opinion, since several similar studies are available in the literature.

It is essential that the authors highlight the innovative elements of their work.

Answer: plantaricin YKX is a new discovered bacteriocin. Although there are many literatures focused on the antibacterial activity of bacteriocin on Staphylococcus aureus. There are few focused on the bacteriocin on the QS system (especially AI-2/LuxS system) of Staphylococcus aureus. We added these description to manuscript (L52-60).

Other points

To my opinion, an introductory paragraph on bacteriocins (activity, applications, etc) should be included in the Introduction.

Answer: thank you, this is amended. The introductory paragraph was deleted in discussion section. L42-52

Discussion of potential application of the plantaricin is missing. Has it been tested in real systems?

Answer:thank you for the comments. We added the potential application of bacteriocins (L42-52) in the manuscript. However, I am sorry, we did not focus on the application. so in this manuscript, we did not test plantaricin YKX on the real food matrix.

Round 2

Reviewer 2 Report

The manuscript has been significantly improved and I think it is now suitable for publication.